# Photochemical alteration of organic carbon draining permafrost soils shifts microbial metabolic pathways and stimulates respiration

Collin P. Ward [1,2], Sarah G. Nalven[3], Byron C. Crump[3], George W. Kling [4] & Rose M. Cory [1]

In sunlit waters, photochemical alteration of dissolved organic carbon (DOC) impacts the microbial respiration of DOC to $CO_2$. This coupled photochemical and biological degradation of DOC is especially critical for carbon budgets in the Arctic, where thawing permafrost soils increase opportunities for DOC oxidation to $CO_2$ in surface waters, thereby reinforcing global warming. Here we show how and why sunlight exposure impacts microbial respiration of DOC draining permafrost soils. Sunlight significantly increases or decreases microbial respiration of DOC depending on whether photo-alteration produces or removes molecules that native microbial communities used prior to light exposure. Using high-resolution chemical and microbial approaches, we show that rates of DOC processing by microbes are likely governed by a combination of the abundance and lability of DOC exported from land to water and produced by photochemical processes, and the capacity and timescale that microbial communities have to adapt to metabolize photo-altered DOC.

[1] Earth and Environmental Sciences, University of Michigan, Ann Arbor, MI 48109-1048, USA. [2] Marine Chemistry and Geochemistry, Woods Hole Oceanographic Institution, Woods Hole, MA 02543-1050, USA. [3] Earth, Ocean, and Atmospheric Sciences, Oregon State University, Corvallis, OR 97331-5503, USA. [4] Department of Ecology and Evolutionary Biology, University of Michigan, Ann Arbor, MI 48109-1048, USA. Correspondence and requests for materials should be addressed to R.M.C. (email: rmcory@umich.edu)

Climate change is rapid in the Arctic, thawing large areas of permafrost[1] that contain nearly half of the world's soil organic carbon[2,3]. As much as 40% of the net land–atmosphere C exchange in the Arctic is mediated by surface waters[4,5], where dissolved organic carbon (DOC) flushed from soils is oxidized to $CO_2$ by sunlight and microbes[6-25]. Photochemical processing of DOC likely supplies about one-third of the $CO_2$ released from Alaskan arctic surface waters[6], by either directly mineralizing DOC to $CO_2$ or indirectly altering DOC chemical composition and, in turn, rates of microbial respiration[13,24]. $CO_2$ emissions from the coupled photochemical and biological degradation of DOC may change as newly thawed DOC from deeper, permafrost soils is exported to surface waters. Permafrost DOC is photochemically labile[13-17], has a different chemical composition compared to DOC currently exported to surface waters from the annually thawed "active layer" of soil[13,18-22], and DOC chemical composition is known to influence the rate of DOC degradation by sunlight and microbes[13,14,19-25].

We know that sunlight and microbes interact to degrade DOC in surface waters, but at present we cannot predict the rate and extent of this degradation either in the dark or in the light[10,11]. This is because the many thousands of organic molecules within soil DOC vary in lability to microbes, and even the more inclusive chemical classes of DOC (e.g., "aromatics" or "aliphatics") may fuel microbial respiration at different rates[25-28]. The general expectation is that smaller (i.e., lower molecular weight), more aliphatic (i.e., higher H/C elemental ratios), and less oxidized (i.e., lower O/C) DOC is easier for microbes to degrade (i.e., more labile) than larger, more aromatic, and more oxidized DOC[29,30]. For example, DOC leached from permafrost across the Arctic is relatively enriched in smaller, more aliphatic and less oxidized DOC compared to DOC draining the upper active, organic soil layer[20-22]. This observation is consistent with findings that permafrost DOC supports relatively higher bacterial growth efficiencies and respiration rates on a per C basis, compared to DOC draining the thawed layer[19-22]. However, when labile molecules comprise too little of the DOC pool, the more abundant but less labile DOC may primarily fuel respiration to $CO_2$[26-28]. In the active, organic layer of arctic soils, the most abundant DOC compounds are supposedly low lability, larger, aromatic, and more oxidized[20-22]. Despite this lower lability, natural microbial communities in arctic soils have the genomic potential to metabolize this more aromatic and more oxidized DOC[31-33], and evidence suggests that these compounds are degraded by soil microbes[13,18,25]. It is likely that microbes degrade DOC spanning a wide range of labilities, with more abundant compounds accounting for the majority of the DOC degraded. Therefore, understanding the effect of sunlight on DOC degradation by microbes requires knowledge of how sunlight alters the pools of both labile and abundant DOC.

Explanations for why sunlight alters rates of DOC degradation by microbes have focused mainly on the photochemical production or loss of small pools of labile DOC, rather than the production or loss of the abundant DOC that may fuel the bulk of respiration. The current model of understanding is that more of the larger, more aromatic, and more oxidized DOC leads to a greater positive effect of sunlight exposure on rates of microbial consumption of DOC because sunlight converts this less labile C to more labile C[34,35]. However, there are some key exceptions to this generalized model. For example, the opposite relationship has been observed for DOC draining Alaskan permafrost soils[13], where permafrost DOC was depleted in larger, more aromatic, and more oxidized DOC compared to DOC draining the active organic layer[22]. Nevertheless, sunlight exposure enhanced microbial respiration of this permafrost DOC by > 40% compared to permafrost DOC held in the dark[13]. In another example, sunlight exposure of aromatic-rich DOC draining the annually thawed soil layer slowed microbial respiration by up to 70%[13,24]. In these Alaskan soils, sunlight exposure of DOC leached from either permafrost or organic layer converted less labile DOC (i.e., larger, more aromatic, and more oxidized) into more labile DOC (i.e., smaller, more aliphatic, and less oxidized)[13,14]. Thus, while photo-exposure of DOC draining permafrost soils in the Arctic may consistently produce labile C, variability in the microbial response to photochemically altered DOC suggests that photo-production of labile DOC is only one factor controlling the effect of sunlight on microbial degradation of DOC.

Here we show that the effect of sunlight on rates of microbial degradation of permafrost DOC depends on photochemical production or removal of the most abundant DOC primarily fueling microbes. We use short-term photochemical experiments, high-resolution mass spectrometry, and measures of microbial activity, community composition, and gene expression to show that in the dark, microbes native to the deep permafrost or surface organic layer soils degraded the DOC that was most abundant in either soil, and sunlight exposure either produced or removed the abundant DOC used by microbes, which induced changes to key metabolic steps taken by the native microbial communities to adapt to and degrade the light-altered DOC. Alteration of permafrost DOC by sunlight to compounds used by microbes results in a two-fold increase in respiration rates, suggesting that when permafrost DOC is exported to sunlit surface waters it can be rapidly respired to $CO_2$.

## Results

**Overview**. Here we explain how and why the photo-alteration of DOC draining the deep permafrost layer stimulates microbial activity[13], but photo-alteration of DOC draining the shallow, annually thawed organic layer suppresses microbial activity[13,24]. We compare the chemical formulas altered by sunlight[14] to the formulas used by microbes (Fig. 1), and relate these changes in DOC chemistry caused by sunlight and microbes to the rates of microbial activity in the light and dark (Fig. 2) and to the shifts in microbial gene expression and community composition (Fig. 3).

**Characterization of DOC consumed by microbes in the dark.** Microbes in each soil layer (the deep permafrost and shallow organic layer) consumed the DOC that was most abundant in that layer. High-resolution mass spectrometry showed that permafrost DOC was enriched in smaller, more aliphatic, and less oxidized formulas compared to organic layer DOC (referred to here as aliphatic-like DOC)[22]. The subset of aliphatic-like formulas initially enriched in permafrost DOC was consumed (Fig. 1a), and supported higher activity rates and growth efficiencies for microbes native to permafrost compared to microbial activity in the organic layer. These findings support the idea that the aliphatic-like DOC is easier for microbes to degrade and yields more energy (i.e., higher lability) than larger, more aromatic, and more oxidized DOC (referred to here as aromatic-like)[29,30]. In contrast, organic layer DOC was enriched in aromatic-like formulas compared to permafrost DOC[22]. Soil microbes native to the organic layer consumed these aromatic-like formulas (Fig. 1b) at lower rates of respiration and growth efficiency compared to rates and efficiencies in permafrost.

There are three plausible explanations why microbes in the organic layer consumed aromatic-like DOC (i.e., lower lability), rather than the aliphatic-like DOC expected to be most labile. First, the concentration of higher-lability, aliphatic-like DOC in the organic layer may have been too low to sustain microbial populations. However, the concentration of aliphatic-like DOC in the organic layer was at least 600 µM C, about three to four times

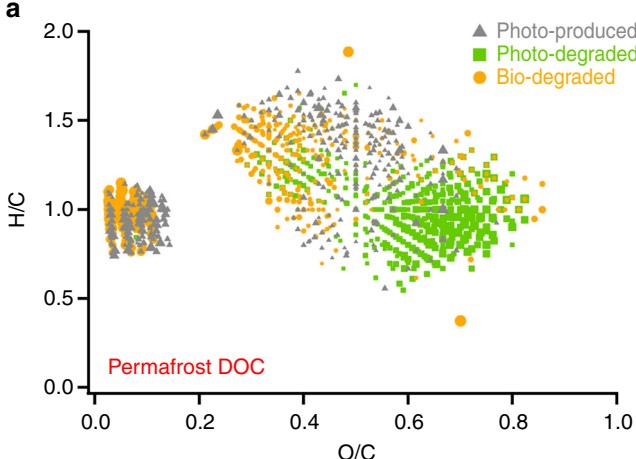

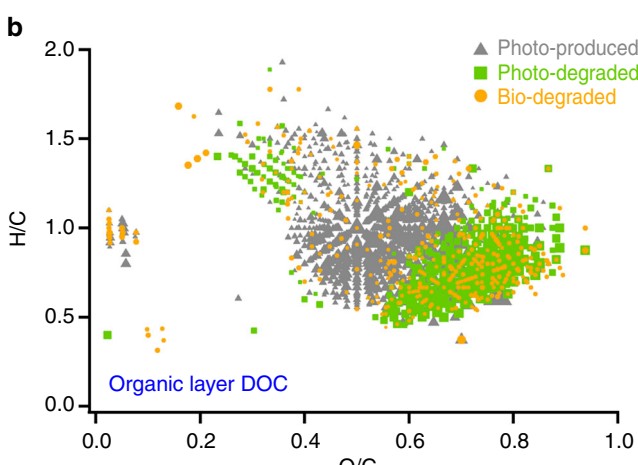

**Fig. 1** Changes to dissolved organic carbon chemical composition by microbes or sunlight. van Krevelen diagrams of formulas in **a** permafrost dissolved organic carbon (DOC) and **b** organic layer DOC that were labile to microbes in the dark (*orange circles*), produced by sunlight (*gray triangles*), or degraded by sunlight (*green squares*). Formulas were categorized as degraded by microbes, produced by sunlight, or degraded by sunlight based on changes in relative formula intensities calculated using the 95% confidence interval of experimental triplicates. The size of the symbol is proportional to the change in relative intensity. Photo-produced and photo-degraded formulas were previously reported[14]

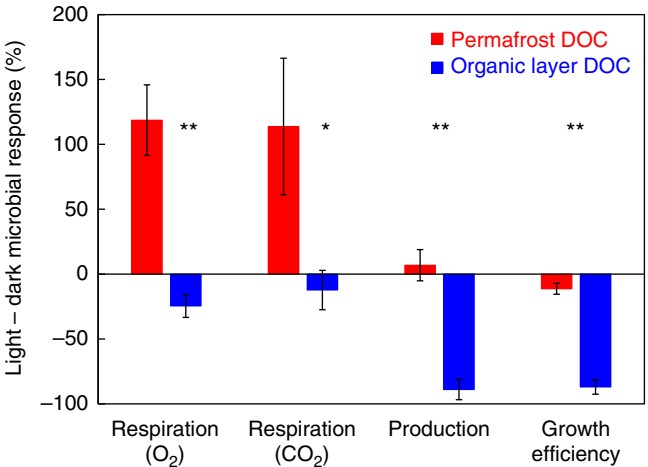

**Fig. 2** Microbial activity response to photo-altered dissolved organic carbon. Light-exposed minus dark-control percent change in microbial respiration ($O_2$ consumption and $CO_2$ production), microbial production, and microbial growth efficiency for permafrost dissolved organic carbon (DOC, *red*) and organic layer DOC (*blue*). *Error bars* indicate ±1 standard error of the mean ($N = 3$). Significance levels reported for an unpaired *t*-test: * = $P < 0.05$, ** = $P < 0.01$

rare in the initial organic layer community to degrade this DOC, especially given the tight coupling between microbial community composition and the degradation of specific types of DOC[23, 24, 37–39]. The relative likelihood of these three explanations can be tested by examining shifts in microbial activities, community compositions, and gene expressions in response to photochemically-altered DOC.

**Characterization of photo-altered DOC consumed by microbes.** Light exposure altered the chemical compositions of permafrost and organic layer DOC, and increased or decreased rates of microbial respiration depending on whether light produced or removed the formulas that fueled native microbes in the dark (Fig. 1a). For permafrost DOC, the main effect of light exposure was to produce more of the same aliphatic-like DOC that permafrost microbes consumed in the dark (Fig. 1a). For example, 26% of photo-produced permafrost DOC formulas had the same exact masses as formulas degraded by permafrost microbes in the dark (Supplementary Fig. 1), and the remaining photo-produced formulas had labile characteristics (i.e., aliphatic-like; Fig. 1a and Supplementary Table 1). Therefore, the effect of sunlight on permafrost DOC was to convert less labile, aromatic-like DOC into compounds labile to the microbes native to permafrost; the lability of these photo-produced compounds was demonstrated by a doubling of respiration rates ($CO_2$ production) compared to dark-controls (Fig. 2 and Supplementary Table 2). In contrast, light exposure of organic layer DOC removed the formulas that were consumed by organic layer microbes in the dark (i.e., removal of aromatic-like DOC; Fig. 1b). For example, 39% of formulas removed by sunlight had the same exact masses as formulas consumed by microbes in the dark (Supplementary Fig. 1), and there was strong overlap in the composition of formulas removed by sunlight and formulas consumed by microbes (Fig. 1b and Supplementary Table 1). The photochemical removal of organic layer DOC used by microbes was consistent with lower microbial respiration and especially production and growth efficiency of light-exposed organic layer DOC compared to dark-controls (Fig. 2 and Supplementary Table 2).

Alternatively, the amount of unsaturated hydrocarbons produced by photo-alteration of organic layer DOC was

greater than the concentration of DOC respired to $CO_2$ or fixed into biomass during the relatively short (5 days) incubation (Supplementary Table 3; based on $^{13}$C-NMR and mass spectrometry showing that 20–30% of the DOC in the organic layer was aliphatic-like, defined as aliphatic DOC resonating from 0 to 60 ppm or aliphatic C with an O/C < 0.5)[29, 30]. These data suggest that microbial use of aliphatic-like DOC in the organic layer was likely not limited by the concentration of this DOC. Alternatively, the second explanation for lack of consumption of aliphatic-like DOC by microbes native to the organic layer is that this DOC may have lower lability than expected. Each molecular formula detected by high-resolution mass spectrometry likely corresponds to many different structural- and stereo-isomers with distinct bonding environments and activation energies for microbes to overcome during metabolism[36]. Thus, there might be critical differences in structure between the aliphatic-like DOC in the organic layer vs. that in permafrost, thereby limiting microbial consumption of this C. Third, microbes with the metabolic capacity to degrade aliphatic-like DOC were either absent or too

insufficient to stimulate the microbial community. A small fraction of the organic layer DOC produced by sunlight could be categorized as unsaturated hydrocarbons (2% of formulas and summed formula intensity, a chemical class expected to stimulate microbial activity)[29, 30]. A similarly small pool of these formulas categorized as unsaturated hydrocarbons was degraded by organic layer microbes (4% of the formula intensity). This result suggests that photo-production of unsaturated hydrocarbons

could have stimulated the activity of a small subset of the microbial community that was genetically equipped to metabolize these compounds. However, the net effect of light exposure on the microbes in the organic layer was suppression (relative to dark-controls), likely resulting from the degradation of the much larger pool of tannin-like formulas that the microbes were consuming (Fig. 1). Others have reported that photo-degradation of DOC in northern humic lakes suppressed microbial activity despite production of labile DOC readily consumed by microbes[40]. Together, findings from this and previous studies suggest that the net effect of photo-altered DOC on microbial activity depends not only on the lability of the photo-products but also on the types of DOC that the microbial communities were equipped to degrade prior to light exposure.

**Effects of DOC photo-alteration on microbial communities.** Photochemical production or removal of formulas fueling native microbes in the dark caused microbial communities to change in both magnitude and direction. The magnitude of community composition change from the initial inoculum in permafrost and organic layer DOC incubations was highest for the treatments that had higher growth rates (i.e., light-exposed permafrost DOC and dark-control organic layer DOC; Fig. 3a). In treatments with lower growth rates (i.e., dark-control permafrost DOC and light-exposed organic layer DOC), the communities changed less over time (Fig. 3a). This suggests that changes in microbial community composition depend on the abundance of DOC that the communities were equipped to degrade. In the permafrost incubations, the large shift in microbial community composition after 4 h, especially for microbes incubated with light-exposed DOC (Fig. 3b), likely reflected the rapid growth of taxa that were adapted to consume newly produced DOC (aliphatic-like DOC produced by light; Fig. 1a). This consumption of labile DOC is consistent with the higher rates of respiration in the light-exposed treatment, where photo-exposure increased the abundance of labile formulas (Fig. 1a) compared to the dark-control (Fig. 2). In the organic layer incubations, shifts in community composition over time were greater for microbes incubated with dark-control DOC (Fig. 3c), likely because there was no shortage of aromatic-like DOC that microbes were equipped to metabolize (Fig. 1b). In contrast, for microbes incubated with light-exposed DOC from the organic layer, the much smaller shift in community composition may be explained by the photo-removal of the aromatic-like DOC most used by this community (Fig. 1b).

The directions of change in community composition were different for permafrost and organic layer communities and for light and dark treatments (directions of *arrows* in Fig. 3b vs. 3c).

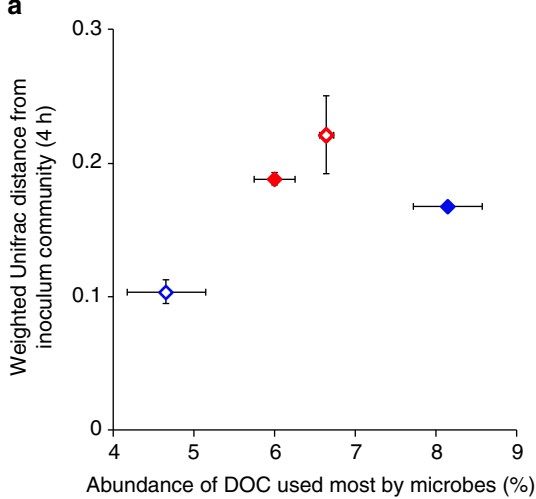

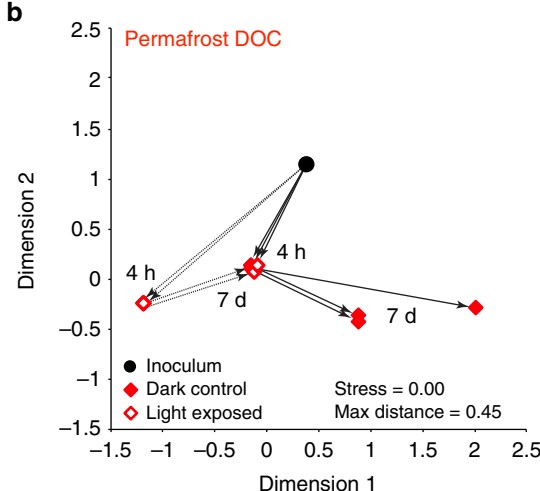

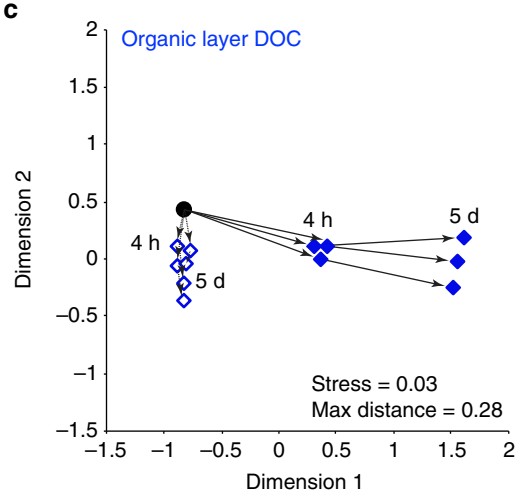

**Fig. 3** Microbial community response to photo-altered dissolved organic carbon. **a** The magnitude of community shift from inoculum to the 4-h time points (weighted Unifrac distance) plotted against the abundance of dissolved organic carbon (DOC) most used by microbes native to each soil layer. DOC most used by microbes was defined following the results in Fig. 1 and Supplementary Table 1: permafrost = aliphatic-like, atomic oxygen to carbon ratio < 0.5, molecular weight < 500 Da; organic layer = aromatic-like, atomic oxygen to carbon ratio > 0.6, molecular weight > 500 Da. *Error bars* indicate ±1 standard error of the mean and in some cases are smaller than the symbol ($N = 2$ or 3). Non-metric multidimensional scaling diagrams showing the magnitude and direction of change in microbial community composition (16S rRNA gene amplicon sequences) among inocula and incubations for **b** permafrost soil leachate, and **c** organic layer soil leachate based on beta-diversity calculations (weighted Unifrac distances for a rarefied data set; 3800 sequences per sample). Note that in **b**, permafrost, the dark 4-h symbols are hidden behind the light, 7-day symbols

In the permafrost layer incubations, after 7 days the microbial community incubated with light-exposed DOC shifted to a community that resembled the dark-control community at 4 h (Fig. 3b), likely reflecting the depletion of sunlight-produced DOC and the growth of taxa that were genetically adapted to consume the less labile dark-control DOC. This result suggests strong overlap in the composition of DOC consumed by microbes between the dark treatment at 4 h and the DOC remaining in the light treatment after 7 days, because DOC chemistry is a principle control on microbial community composition[23, 24, 37–39]. Such an overlap in composition is consistent with our suggestion that photo-exposure of permafrost DOC increased the abundance of compounds fueling microbes in the dark. In the organic layer incubations, after 5 days the community in the light-exposed treatment had changed little and, unlike in the permafrost incubations, this change was to a community that was different than the dark-control (Fig. 3c), suggesting that sunlight removed the compounds that were consumed by the dark-control community (Fig. 1b). These shifts in community composition are consistent with the reduction in activity levels in the light-exposed treatment (Fig. 2), and suggest that populations with different metabolic potentials must be required to degrade the new, photo-produced aliphatic-like DOC. Overall, the magnitude and direction of these shifts in permafrost and organic layers likely reflect the adaptation of microbial communities to the chemistry of the organic matter they are consuming, both over the short term (hours) due to metabolic (physiological) responses of individual cells, and over the long term (days, as shown in the 16S results of Fig. 3) due to selection for microbial populations (species) with better-suited metabolic machinery.

The suggestion that microbes adapt their metabolic machinery in response to photo-altered DOC was tested using metatranscriptomic measurements. Four hours into the incubation the expression of genes coding for the degradation of aromatic molecules (KEGG Tier IV Category) was significantly lower for the organic layer community incubated with photo-altered DOC compared to the dark-control community, regardless of whether expression was normalized to all KEGG gene expression or to Metabolism gene expression (KEGG Tier II Category; paired t-test, $P \leq 0.05$)[41]. Moreover, 15 of the 16 differentially expressed aromatic-degradation genes had lower expression in the light treatment than in the dark-control[41]. This result suggests that microbes incubated with photo-altered organic layer DOC re-tooled their metabolic machinery to degrade the labile, aliphatic-like DOC that was produced in the light (Fig. 1b). Consistent with this interpretation, prior to light exposure of DOC the metabolic pathways of microbes native to the organic layer were more focused on consuming lower lability aromatic-like DOC rather than the less abundant aliphatic-like DOC (Fig. 1b and Supplementary Table 1). Over time, this initial metabolic response would lead to a competitive advantage for populations with the metabolic potential to degrade the aliphatic-like DOC that was produced by sunlight (Fig. 1b). Together, the differential gene expression and changes in community composition (Fig. 3b, c) suggest that sunlight exposure either produced (in the permafrost layer) or removed (in the organic layer) the abundant DOC that the microbial community was equipped to degrade, thereby inducing changes to key metabolic pathways used by the native microbial communities to consume DOC.

**Synthesis of microbial responses to photo-altered DOC**. The results of this study provide a mechanistic interpretation for the reported suppression of microbial activity when DOC from the upper, organic soil layer in the Alaskan tundra is exposed to sunlight (Fig. 2)[13, 24]. The net negative effect of light exposure on

microbial activity and consumption of organic layer DOC is likely due to photo-removal of DOC that organic-mat microbial communities were metabolically equipped to degrade (i.e., aromatic-like DOC; Fig. 1b). In addition, our findings may explain reported lags in rates of microbial respiration (i.e., DOC consumption) of photo-altered DOC, that, after longer incubation (i.e., weeks)[24], eventually reach or exceed rates of respiration for dark-controls[24, 42, 43]. Photochemical alteration converts larger, more aromatic, more oxidized DOC into smaller, more aliphatic, less oxidized DOC that can be consumed faster and with greater efficiencies in the presence of a microbial community adapted to degrade this DOC (Fig. 1). Thus, the negative effect of light on microbial processing of organic layer DOC may be temporary given that aquatic microbial communities can adapt to changes in DOC chemical composition on timescales of weeks (Fig. 3)[13, 23, 24, 44]. In the headwater stream draining the organic layer of soils studied here, we consistently observe a net positive effect of sunlight exposure on rates of microbial DOC respiration[45]. Given that the DOC in this headwater stream is similar in chemical composition to the DOC leached from the organic layer in this study[22, 45], the difference in the effect of light on microbial activity may be the time that microbial communities in the stream have to adapt to photo-altered DOC.

An alternative to the explanation that microbial communities need time to adapt to photo-altered organic layer DOC is that photochemically produced reactive oxygen species (ROS) that can be harmful to microbes suppressed the consumption of photo-altered DOC. Photochemical production of ROS has been proposed to account for the lag in microbial growth or respiration from photo-altered DOC given that ROS decay over time following light exposure[46]. However, our experimental design likely minimized any direct effect from ROS on microbes because the light exposure portion of our experiment was conducted without microbes (Supplementary Table 5), and there was a period of about 12 h between light exposure and addition of inoculum to organic layer and permafrost DOC treatments. Thus, photochemically produced ROS in the light-exposed DOC may have decayed to dark-control levels prior to addition of the bacterial inoculum by reacting with DOC or other constituents in the soil waters[47]. Furthermore, others have concluded that the effects of light exposure of DOC on microbial growth rates were more likely due to changes in DOC composition than to harmful effects of ROS on microbes[42]. While it is possible that photochemical production of ROS alters microbial communities and their activities, both our experimental design and evidence from the literature suggest that the primary control on the microbial response observed in this study is the photochemical alteration to DOC chemical composition.

**Implications for DOC fate in the Arctic**. Our findings on the controls of coupled photochemical–biological processing of DOC likely apply to pan-arctic sunlit waters. This is because most surface waters of the Arctic are shallow and unshaded, DOC is the primary light absorbing constituent[6, 48, 49], and DOC draining permafrost soils in many arctic regions has a high susceptibility to photo-degradation[6–9, 12–17, 24, 25]. Over a wide range of Alaskan Arctic soil types, representative of pan-arctic soil types[50], the composition of permafrost and organic layer DOC produced and removed by sunlight followed a similar pattern; larger, more aromatic-like formulas were converted into smaller, more aliphatic-like formulas (Supplementary Tables 3 and 4 and Supplementary Fig. 3). Furthermore, across the six largest rivers of the pan-arctic watershed, DOC chemical composition was one of the strongest predictors of microbial community

composition[38], and together DOC chemical composition and microbial community composition drive rates of DOC consumption by microbes. These results suggest that rates of microbial processing of DOC in pan-arctic inland waters are controlled by the abundance of more labile, smaller, more aliphatic, and less oxidized DOC exported from land to water or produced photochemically, and the capacity and timescale that microbial communities have to adapt to and then degrade this photo-altered DOC.

Given the consistently large impact of sunlight on microbial respiration of permafrost DOC in this and previous work (i.e., increases of > 40%; Fig. 2)[13], the coupled photochemical and biological degradation of permafrost DOC may be an increasingly important component of the arctic C budget as the climate warms. Independent of the rates of permafrost degradation and of the future water balance of northern basins, all newly thawed soils will experience leaching and loss of DOC via drainage into surface waters where this DOC, which is susceptible to both photochemical and biological degradation, will be exposed to sunlight[51, 52]. Photochemical alteration of DOC draining from permafrost soils into small streams and ponds may be most often limited by the amount of light available[45]. Earlier ice-out on lakes in a warmer world could decrease light limitation of photochemistry by increasing UV exposure. This is because solar radiation is highest and clear-sky days are more common in early summer compared to mid-summer and fall[6]. We suggest that a large fraction of permafrost DOC may be rapidly converted to $CO_2$ or microbial biomass when exported to surface waters due to the rapid photo-production of DOC used by microbes.

## Methods

**Experimental design.** Thawed surface soil and frozen deep soil (permafrost) were sampled from three replicate pits in moist acidic tussock tundra within the Imnavait Creek watershed on the North Slope of Alaska (68.62° N, 149.29° W). Deionized water was added to soil from two layers of each pit, the annually thawed, shallow organic layer (5–15 cm depth) and the deeper permafrost layer (95–105 cm), in order to generate the triplicate samples used in each experiment[22]. The dissolved organic carbon fraction of the leachates was isolated using 0.45 μm filters (Geotech Environmental Equipment, Inc.)[22]. Organic and permafrost layer DOC from each pit was placed in UV-transparent Whirl-Pak bags (Nasco, Inc.) and exposed to natural sunlight for 24 h alongside dark-controls[13, 14]. Light-exposed and dark-control DOC from each soil layer was incubated with an inoculum of native microbial communities at 6–7 °C. The inoculum was composed of leachate filtered through a Whatman GF/C (nominal pore size 1.2 μm) and added to comprise 20% of the sample volume. We tested the effect of UV exposure and filtering with GF/F filters (nominal pore size 0.7 μm) and sterile 0.2 μm filters on carryover of bacteria into sample incubations by measuring bacterial production on filtered water compared to whole water controls (Supplementary Table 5). Results indicate that even GF/F filters reduced bacterial contamination (measured as bacterial production) consistently in dark minus control samples (93 ± 2%, mean ± 1 SD), and UV exposure in the light treatment plus filtering reduced production even more (99 ± 1%, mean ± 1 SD; Supplementary Table 5). The effect of sunlight exposure of DOC on microbial community composition and gene expression was assessed 4 h into the incubation. At the end of the incubation (5–7 days), microbial activity (i.e., respiration, production, and growth efficiency) and community composition were quantified[6, 13, 22]. The chemical composition of the initial DOC (i.e., not exposed to sunlight) consumed by microbes was characterized using high-resolution Fourier transform-ion cyclotron resonance mass spectrometry (FT-ICR MS[14, 22, 27, 42]). The relatively short incubation times were chosen to ensure the detection of a change in DOC chemical composition, microbial activity, and microbial community composition, while minimizing the amount of time the DOC and microbes spent in a bottle (i.e., bottle effects).

**Quantifying responses of microbes to DOC.** Measurements of microbial respiration and production have been previously described in detail[6, 13, 22]. Briefly, respiration was quantified as dissolved inorganic carbon production (AS-C3 DIC Analyzer; Apollo SciTech, Inc.) or dissolved oxygen consumption (membrane-inlet mass spectrometry, Bay Instruments)[53] compared to killed controls (1% $H_gCl_2$). Microbial production was quantified as $^{14}C$-labeled L-leucine incorporation into the cold trichloroacetic acid (TCA)-insoluble fraction of macromolecules in viable samples compared to a TCA-killed control. Microbial growth efficiency was calculated as production divided by the sum of production and respiration.

Changes in microbial community composition during incubation with light-exposed or dark-control organic and permafrost layer DOC (Fig. 3 and Supplementary Fig. 4) were quantified using amplicon Illumina sequencing (MiSeq 2 × 150 bp paired-end) of the V4 region of bacterial 16S rRNA genes[39, 54]. Amplicon sequences were paired using make.contigs (MOTHUR v.1.32.1)[55], and converted to QIIME format with split.groups from MOTHUR and add_qiime_labels.py from QIIME[54]. Sequences were quality filtered with an expected error rate of 0.5, dereplicated (derep_fulllength), and abundance sorted (sortbysize) using USEARCH (v.7.0.1001_i86linux64)[56]. Singleton sequences were removed, and reads were clustered (cluster_otus) at 97% similarity. Chimeras were removed with the de novo chimera check inherent in the cluster_otus, and with reference-based chimera filtering (uchime_ref) using the Gold Database (www.genomesonline.org) as reference. Reads (including singletons) were subsequently mapped back to the operational taxonomic unites (OTUs) using UPARSE (usearch_global). Taxonomy of the representative sequences was assigned in QIIME (assign_taxonomy.py) using the RDP classifier trained to the SILVA database (v.111 database clustered to 97% OTUs). Patterns in beta-diversity, calculated as weighted Unifrac distance[57], were based on a rarefied OTU table (3800 sequences per sample) and displayed in non-metric multidimensional scaling diagrams[58] using PRIMER-E software (V 7.0) to show the magnitude and direction of change in microbial community composition.

Metatranscriptome sequences were generated from RNA samples filtered after 4 h incubation onto 0.22 μm filters (Supor; Pall Corp.), preserved with RNA*later* (Qiagen), and extracted and purified[59]. Ribosomal RNA removal, cDNA synthesis, and Illumina HiSeq sequencing were performed at the Joint Genome Institute (JGI) in Walnut Creek, CA, with either standard or low-input RNASeq protocols, using Ribo-Zero rRNA Removal Kits for Bacteria (Epicentre), and Truseq Stranded RNA LT kits (Illumina). RNA sequences were quality-controlled using BBDuk and BBMap, and assembled using MEGAHIT[60]. Coding sequences (CDS) were annotated to the KEGG database[61] and a custom phylogenetic database[62]. Quality controlled reads were mapped to CDS using Bowtie2[63], and counts, CDS lengths, and alignment lengths were extracted with SAMtools[64]. Counts per CDS were normalized to transcripts per million (TPM)[65]. Transcript abundances of genes within the KEGG Degradation of Aromatic Compounds Category (Tier IV) were reported as percentages of total Metabolism (KEGG Tier II Category).

**Characterizing DOC consumed by microbes.** The chemical composition of permafrost or organic layer DOC consumed by native microbes in each layer was characterized using high-resolution FT-ICR MS[14, 22]. Pre- and post-incubated DOC was extracted using PPL solid-phase (SPE) to remove impurities in preparation for FT-ICR MS analysis[66]. DOC recovery for all samples, including the photochemical experiments described below, ranged from 48 to 75% and averaged 63 ± 1% ( ± 1 SE). Methanol SPE eluates were diluted to ~50 mg C per L prior to introduction to the electrospray ionization source of a 12T Bruker SolariX FT-ICR mass spectrometer. All spectra were acquired in negative mode. Formula assignment criteria have previously been described in detail[22]. Formulas were categorized as consumed by or resistant to microbes if their intensity decreased or remained unchanged after incubation with the native microbial community, respectively. The 95% confidence intervals calculated across experimental replicates were used to determine if a change in formula intensity after incubation with microbes was significantly greater than zero ($N = 3$).

**Characterizing DOC altered by sunlight.** Photochemical alterations of DOC leached from the thawed and permafrost layer of three dominant soil types on the North Slope of Alaska were characterized using FT-ICR MS. Chemical attributes of waters leached from all soils are provided in Supplementary Table 3. Photochemical changes to the chemical composition of DOC leached from Imnavait moist acidic tundra (Fig. 1) were previously reported[14]. Photochemical changes to the chemical composition of DOC leached from the other four soil types were determined following previously described protocols[14, 22]. Filtered leachates (< 0.45 μm) were transferred to UV-transparent Whirl-Pak bags (Nasco, Inc.) and exposed to 18 h of simulated sunlight at 20 °C alongside dark-controls (Atlas Suntest XLS+). The simulated sunlight had a similar spectral shape, but was 2.4-fold more intense than average June sunlight at Toolik Field Station, AK (Supplementary Fig. 2). Therefore, 18 h of simulated sunlight is equivalent to approximately 2 days of natural light in June at the field station. Formulas were categorized as photo-degraded or photo-produced if their intensity decreased or increased after light exposure, respectively. The 95% confidence intervals calculated across experimental replicates were used to determine if a change in formula intensity after light exposure was significantly greater than zero. Experimental replicates were not available for Toolik tussock permafrost DOC. Therefore, a change in peak intensity was considered significant if the intensity change was > 20% after sunlight exposure, which is equivalent to twice the coefficient of variation of instrumental replicates. Shifts in compound class distributions for photo-degraded and photo-produced formulas across all soil types are presented in Supplementary Table 4 and Supplementary Fig. 3.

**Data availability.** 16S rRNA gene amplicon sequences are deposited in the NCBI Sequence Read Archive (SRA) under the bioproject accession number PRJNA356108. Metatranscriptome sequences and assembled contigs are publicly

available via IMG under GOLD study ID Gs0114298. All other data presented in this study have been made publicly available online within the Arctic Long Term Ecological Research database.

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

## Acknowledgements

We thank J. Dobkowski, K. Harrold, L. Treibergs, M. Stuart, M. Findley, A. Clinger, S. Michael, and researchers, technicians, and support staff of the Toolik Lake Arctic LTER and Toolik Lake Field station for assistance. Thanks to K. Roscioli and M. Tfaily for assisting with the mass spectrometry analysis, which was performed using EMSL, a DOE Office of Science User Facility sponsored by the Office of BER and located at PNNL. Funding for this work was provided by NSF grants OPP 1023270, 1022876, CAREER 1351745, DEB 1147378, 1347042, 0639790, 1147336, 1026843, PLR 1504006, DOE-JGI-CSP 1782, and the Camille and Henry Dreyfus Foundation Postdoctoral Program in Environmental Chemistry. The data set provided by the Toolik Field Station Environmental Data Center (Supplementary Fig. 2) was supported by NSF grants 455541 and 1048361.

## Author contributions

C.P.W., B.C.C., G.W.K., and R.M.C. designed the study, collected the samples and performed the experimental manipulations. C.P.W. and S.G.N. analyzed the data with input from B.C.C., G.W.K., and R.M.C. The manuscript was written by C.P.W., B.C.C., G.W.K., and R.M.C with input from S.G.N.

## Additional information

**Competing interests:** The authors declare no competing financial interests.

