## [Peer Review File · Nature Communications]

Reviewers' comments:

Reviewer #1 (Remarks to the Author):

Ward et al. report on dissolved organic matter (DOM) dynamics in the freshwater systems of the high Arctic, comparing the responses of bacterial communities native to deep permafrost to those the more active surface layer. They find that each of these communities are adapted to the more abundant DOM component in their environment. They also found that irradiation of the DOM as occurs in surface waters enriched the components of DOM utilized by the permafrost community but depleted DOM components utilized by the surface layer community. The coupling of these two results provides a novel explanation for why pre-irradiation of DOM enhances activity in the permafrost community but has a negative effect on the surface community.

This is an important result to the field as it helps explain why irradiation can have conflicting results on the bio-lability of DOM, an observation that has puzzled biogeochemists for some time. It also shows how the combination of FT-ICR-MS analysis of dissolved organic matter and genomic analysis of the microbial community can bring new insights into biogeochemical processes important to global carbon dynamics. Ward et al. do a good job at laying out the basic results but a number of technical details are not adequately explained making this version less accessible to the specialist in the field (the target audience of the journal). I expect that the authors should be able to clarify these points in revision. Hopefully, the following comments will be useful in improving the presentation.

The point I found that most needed clarification was the presentation of the FT-ICR-MS results to address the point of what changes occur in the abundance of different DOM molecular components. For most of the text, the results are discussed in terms of detectable changes in formulas, that being whether an increase or decrease occurred at the 95% CI. The frequency of such changes in different molecule classes was tracked relative to microbial activity, as described on page 8:

"Photochemical production or removal of formulas fueling native microbes in the dark caused microbial communities to change. "

While the number of formulas that change is a relevant metric, bacteria, shall we say, "don't live on formulas alone". Just as important as whether a change has occurred is the net increase or decrease in mass of the component. At points in the mss it seems that the authors are interpreting frequency of change results as also a change in mass concentration, i.e.

On Page 6 -

"... the amount of higher-lability, aliphatic-like DOC in the organic mat may have been too low to sustain microbial populations. This explanation is unlikely because $22 \pm 1\%$ of the initial organic mat DOC could be classified as labile (aliphatic-like DOC with an O/C less than 0.5), which was larger than the $8 \pm 2\%$ of the DOC pool respired to CO₂ or fixed into biomass during the relatively short (5 d) incubation. "

Here the frequency of formula types is directly compared with a mass measurement of DOC, but those are two different types of information. Only relying on the frequency of change could bias the interpretation. For example, in ref. 14, photoproducted and photodegraded formulas are discussed and a Van Krevelen diagram with these formulas (apparently same as Figure 1) are shown as their Fig. 3. However, the earlier figure additionally indicates the relative change in the mass intensity of each formula through the size of the symbol. From this it is clear that not only were there fewer of the unsaturated hydrocarbon formulas that one might presume to be highly labile but also that irradiation was relatively unproductive of these formulas in contrast to the light treatment of permafrost DOM. The lack of production could have been a factor in the lack of a positive response of the mat community to irradiated DOM. This is offered as an example, the important point is that I don't think that mass (intensity) information should be left out of this

presentation.

Some other points that need clarifications:

1) The description of the inocula for the incubations is quite vague, presently are just described as "native" communities. I see from other publications that inoculum has been taken as the whole water from the same leachate filtered for DOC, indicate if this was also the case. How closely do these communities resemble those present in surface waters –e.g. how much does the composition change once the community is exposed to sunlight?

2) The Methods never actually state the sample location and dates for the results shown in Fig. 1-3. As mentioned, Fig. 1 seems basically identical to Fig. 3 in ref 14 for the photo-produced and -degraded, suggesting the location is the same, Imnavait moist acidic tundra. Please state explicitly. Are all the results from a single leaching? Are the triplicate replicates from three separate experiments or are they measurement replicates conducted at the same time?

3) The photochemical experiment was apparently conducted in natural sunlight (24 hours) for the Imnavait moist tundra, but a solar simulator Suntest XLS+ was used for the other four types. 18 hours in the simulator is described as comparable to 24 hours natural sunlight at the location which I find surprising. Although the spectra shown in Fig. S2 are similar, the solar spectrum looks like mid-day clear sky. The average over 24 h would be much lower (even in June, UV content is very low for the portion of the day the sun is near the horizon). The figure legend cites reference 3 as the source for the natural spectrum, but that reference does not have any data on solar irradiance. Given how small is the Suntest XLS+ exposure chamber the authors should explain how they were able to configure a 45 cm distance between the lamp and the sample tubes. The authors should keep in mind that borosilicate glass generally filters out the shortest wavelength part of the UVB (<315 nm), probably of minor importance in photobleaching but should be accounted for in any modeling calculations.

Minor Comments

1) Abstract "experimental manipulations of soils"
Change "soils" to "soil leachates"

2) Results – the first paragraph states some important findings, however every sentence has a reference to another report. I found it difficult to understand what was being presented as new data in this report and what was found previously.

3) "This explanation is unlikely because $22 \pm 1\%$ of the initial organic mat DOC could be classified as labile (aliphatic-like DOC with an O/C less than 0.5)²¹, which was larger than the $8 \pm 2\%$ of the DOC pool respired to CO₂ or fixed into biomass during the relatively short (5 d) incubation."

In addition to the point about abundance already mentioned, I found it difficult to follow the reasoning here. The assimilation rate of a DOC class will depend more on its concentration than on its percentage, and the percent of the total DOC pool respired or fixed will depend on the abundance of bacteria as well as the lability of the DOC.

4) "For example, 26% of photo-produced permafrost DOC formulas had the same exact masses as formulas degraded by permafrost microbes in the dark, and the remaining photo-produced formulas had labile characteristics (i.e., aliphatic-like; Fig. 1A; Table S1). "

and

"For example, 39% of formulas removed by sunlight had the same exact masses as formulas consumed by microbes in the dark, and there was strong overlap in the composition of formulas

removed by sunlight and formulas consumed by microbes (Fig. 1B, Table S1)."

These are important points, but neither Fig. 1 nor Table S1 show the formulas that were both degraded by microbes and photo- produced/removed. In the figure it looks like each formula can have only one symbol type.

5) The description of community change seems to be only in terms of Unifrac distance and there is no interpretation of the directional change in the MDS space. Suggest either explaining what are the axes and what the directions mean or just presenting the distances in an unscaled diagram.

In any case, I am not sure what new information is being added. If activity of a community is stimulated, composition change would be expected since some strains will be better able to use the available material than others. The significance of the overlap in the dark treatment at four hours and the light treatment at 7 days is not clear since the dark treatment has also shifted by about that much distance in 7 days.

6) "Thus, the negative effect of light on microbial processing of organic mat DOC may be temporary given that aquatic microbial communities can adapt to changes in DOC chemical composition on timescales of hours to days (Fig. 3)13,22,23,41."

However, the composition of the organic mat community in the light treatment did not change over 5 days.

7) " However, addition of hydrogen peroxide, a harmful ROS produced by DOC in sunlit surface waters, at a greater concentration than was produced during photo-degradation of permafrost and organic mat DOC14, showed no effect on microbial growth39."

What was the concentration of the added H2O2 and where is it shown that there was no effect? Ref. 39 deals with Lake Superior bacterioplankton.

8)Methods- "UV exposure during our experiments killed bacteria and reduced production by >99% (Table S5)."

Production is not a definitive measure that the bacteria are killed, leucine incorporation in incubations of bacterioplankton has been observed to largely recover during the night even after substantial inhibition occurs during the day.

8) Figure 1 Legend – misspelling "catergorized". States that formulas produced by or degraded by sunlight are shown in ref 13, but Table S1 states that they are shown in ref 14.

9) Table 4.S1 "All data previously reported in Ward and Cory, In Review."
Update the reference

Respectfully submitted,

Patrick Neale
Edgewater, MD

Reviewer #2 (Remarks to the Author):

General Comments

The authors present an interesting study focused on carbon-cycle dynamics in waters draining

Arctic permafrost landscapes in northern Alaska. Specifically, this work examines the effects of photo-chemical modification of DOC composition and microbial respiration, with implications for C release from soils and surface waters to the atmosphere. The authors build upon prior work linking the physical effects of photo-oxidation and biological effects of microbial degradation on DOC production and turnover. Their findings show that sunlight can have both positive or negative impacts on microbial activity and respiration depending on whether dominant DOM pool is produced or removed via photo-oxidation. The authors used a combination of complimentary methods to disentangle mechanisms and test hypotheses, including FTICR-MS (for DOM composition), microbial activity and composition, and gene expression over the course of a 5-7 day incubation. The FTICR-MS approach used appropriate methods, and interpretation of results relied heavily on methods/results reported by Cory et al. (2013 PNAS). The manuscript is well written and advances our understanding of C dynamics in this critical region. The FTICR-MS approach used appropriate methods, and interpretation of results relied heavily on methods/results reported by Cory et al. (2013 PNAS). Most of my comments are relatively minor below.

Specific comments

1. Title: While it's true that some pathways "stimulate respiration", other findings show a reduction in rates. Consider rephrasing title for accuracy.
2. Page 3, Paragraph 2: Omit "soils" after "DOC leached from permafrost". In the literature, "permafrost soils" can include both active layer and perennially frozen soils (see Hugelius et al. 2014).
3. Page 4, First line – Since "active layer soils" are defined above, I suggest using "active layer" throughout the manuscript. Further, the active layer thaws and refreezes annually, and is only "thawed" a couple months out of the year.
4. Page 4, Last line – "Organic-horizon" or "Organic layer" are preferred terminology compared to "Organic mat".
5. Page 6, Last paragraph – I like the discussion DOM lability in this section, but it could use a bit more detail/context. Could you add a sentence briefly describing the approach used by Ward & Cory (2015) to categorize lability? Why was the incubation only run for 5-7 days (and why the variability of 5-7?).
6. Page 13 – I recommend adding a sentence or two to describe how soils were sampled and how leachings were conducted.
7. Figure 2 – What statistical test was used here to determine differences across means? I don't see data analyses described anywhere in Methods.
8. Figure 3 – While I recognize space is limited, there is little to no text that describes the multidimensional scaling method and details of the results. I recommend adding a sentence or two somewhere to cover this in more detail.

Reviewer #3 (Remarks to the Author):

The authors Ward et al., present a manuscript titled "Photochemical alteration of dissolved organic carbon draining permafrost soils shifts microbial metabolic pathways and stimulates respiration". Overall, the manuscript is well presented and written, but I feel the following concerns need to be addressed before publication.

Results and Discussion

In the first line the authors introduce the term "organic mat". I am not sure exactly what this is and how it differs from active layer. There is a line in the methods that indicates it is the seasonally thawed layer. I would move this information to the results section as the reader sees this first in the organization of the manuscript. Also, please distinguish the difference between organic mat and active layer, it may be a common term, but one which I have not encountered

before.

In the Results and discussion it is sometimes hard to distinguish findings from this study versus references to the literature. For example, in the same sentence there is reference to a figure and other literature. This sometimes makes it difficult to decide if the results are novel to this study or verify results of other studies. I would try to split these with statements like, similar to whoever et al., we found..... Try to highlight the novel findings of this study.

Carful with the interpretation of the data. The authors state "microbes adapt their metabolic machinery". However, the data also shows that there were taxonomic shifts in the community. Adapting their metabolic machinery implies that a single organism is shifting its own gene expression in response to an environmental change. Instead, your data seems to more strongly imply that the environmental change enriches different populations with different metabolic potentials. I think this is an important distinction that should be called out.

Methods

As mentioned previously I think the introduction to the methods needs to be in the results to introduce terms and concepts to the reader early.

I would suggest that always in the phrase "by always >90%" should be replaced by consistently. While I understand that there are limits on sampling and sequencing etc. I think there needs to be some discussion to the fact that transcriptomes were generated at 4 hours and respiration data at 5-7 days. Also why organic mats were incubated 5 days versus 7 days for permafrost. While I don't see why this would affect the overall conclusions, it seems rather arbitrary. If it was based on some type of data or measurement I would make this abundantly clear, or else add some caveats to the data interpretation.

Changes in microbial composition. There needs to be more information of the community composition data. There is no information on what OTU definition was used in the calculations. Also tools in Quiime Mothur and Usesrach is not sufficient. Which tools, default parameters? Please expand. Someone who wanted to recapitulate your study would have no way to do so with the information you have provided.

I also could not find a statement as to the public availability of the datasets. Please make sure that the datasets are available in a public repository before publication.

Red italics = response to reviewer comment

Black italics = revisions to text

Reviewer #1 (Remarks to the Author):

Ward et al. report on dissolved organic matter (DOM) dynamics in the freshwater systems of the high Arctic, comparing the responses of bacterial communities native to deep permafrost to those the more active surface layer. They find that each of these communities are adapted to the more abundant DOM component in their environment. They also found that irradiation of the DOM as occurs in surface waters enriched the components of DOM utilized by the permafrost community but depleted DOM components utilized by the surface layer community. The coupling of these two results provides a novel explanation for why pre-irradiation of DOM enhances activity in the permafrost community but has a negative effect on the surface community.

This is an important result to the field as it helps explain why irradiation can have conflicting results on the bio-lability of DOM, an observation that has puzzled biogeochemists for some time. It also shows how the combination of FT-ICR-MS analysis of dissolved organic matter and genomic analysis of the microbial community can bring new insights into biogeochemical processes important to global carbon dynamics. Ward et al. do a good job at laying out the basic results but a number of technical details are not adequately explained making this version less accessible to the specialist in the field (the target audience of the journal). I expect that the authors should be able to clarify these points in revision. Hopefully, the following comments will be useful in improving the presentation.

The point I found that most needed clarification was the presentation of the FT-ICR-MS results to address the point of what changes occur in the abundance of different DOM molecular components. For most of the text, the results are discussed in terms of detectable changes in formulas, that being whether an increase or decrease occurred at the 95% CI. The frequency of such changes in different molecule classes was tracked relative to microbial activity, as described on page 8:

"Photochemical production or removal of formulas fueling native microbes in the dark caused microbial communities to change. "

While the number of formulas that change is a relevant metric, bacteria, shall we say, "don't live on formulas alone". Just as important as whether a change has occurred is the net increase or decrease in mass of the component. At points in the mss it seems that the authors are interpreting frequency of change results as also a change in mass concentration, i.e.

On Page 6 -

"... the amount of higher-lability, aliphatic-like DOC in the organic mat may have been too low to sustain microbial populations. This explanation is unlikely because $22 \pm 1\%$ of the initial organic mat DOC could be classified as labile (aliphatic-like DOC with an O/C less than 0.5), which was larger than the $8 \pm 2\%$ of the DOC pool respired to CO₂ or fixed into biomass during the relatively short (5 d) incubation. "

Here the frequency of formula types is directly compared with a mass measurement of DOC, but those are two different types of information. Only relying on the frequency of change could bias

the interpretation. For example, in ref. 14, photoproducted and photodegraded formulas are discussed and a Van Krevelen diagram with these formulas (apparently same as Figure 1) are shown as their Fig. 3. However, the earlier figure additionally indicates the relative change in the mass intensity of each formula through the size of the symbol. From this it is clear that not only were there fewer of the unsaturated hydrocarbon formulas that one might presume to be highly labile but also that irradiation was relatively unproductive of these formulas in contrast to the light treatment of permafrost DOM. The lack of production could have been a factor in the lack of a positive response of the mat community to irradiated DOM. This is offered as an example, the important point is that I don't think that mass (intensity) information should be left out of this presentation.

Yes, on page 6 of the original manuscript we did discuss the mass spec results in the context of the amount of DOC (the mass of C or the concentration) that was respired. However, we agree with the reviewer that including the change in formula intensity (i.e. the intensity corresponding to the mass of each ion detected) strengthens the interpretation of the results. We updated the van Krevelen diagrams in Fig. 1 of the main text such that the size of the symbol corresponds to the magnitude in intensity change for each formula mass detected.

We revised the text on pages 8-9 to clarify that the concentration of organic layer DOC categorized as labile was higher than the amount of DOC respired or fixed into biomass during the incubation. This result suggests we can rule out the possibility that the organic layer microbial community consumed less-labile, aromatic-like DOC because it was limited by the concentration of labile, aliphatic-like DOC available to the community.

“There are three plausible explanations why microbes in the organic layer consumed aromatic-like DOC (i.e., lower lability), rather than the aliphatic-like DOC expected to be most labile. First, the concentration of higher-lability, aliphatic-like DOC in the organic layer may have been too low to sustain microbial populations. However, the concentration of aliphatic-like DOC in the organic layer was at least 600 μM C, about three to four times greater than the concentration of DOC respired to CO_2 or fixed into biomass during the relatively short (5 d) incubation (Table S3; based on ^{13}C -NMR and mass spectrometry showing that 20-30 % of the DOC in the organic layer was aliphatic-like, defined as aliphatic DOC resonating from 0-60 ppm or aliphatic C with an $\text{O}/\text{C} < 0.5$)^{29,30}. These data suggest that microbial use of aliphatic-like DOC in the organic layer may have been limited by the lability, not the concentration, of this DOC.”

We agree that there are two potential explanations of the negative response of the microbial community to photo-altered organic layer DOC: (1) sunlight removed formulas that the microbes preferred to use (emphasized in our original manuscript), or (2) the amount of unsaturated hydrocarbons produced by sunlight was insufficient to stimulate the microbial community (we added text to discuss this interpretation in our revised manuscript). The overall response of microbes to photo-altered DOC could be due to a combination of (1) and (2). Although there is more evidence in favor of the first explanation that sunlight removed formulas that the microbes preferred to use, we added text on pages 8-9 to address this reviewer's point:

“Alternatively, the amount of unsaturated hydrocarbons produced by photo-alteration of organic layer DOC was insufficient to stimulate the microbial community. A small fraction of the organic layer DOC produced by sunlight could be categorized as unsaturated hydrocarbons (2% of formulas and summed formula intensity, a chemical class expected to stimulate microbial activity)^{29,30}. A similarly small pool of these formulas categorized as unsaturated hydrocarbons was degraded by organic layer microbes (4% of the formula intensity). This result suggests that photo-production of unsaturated hydrocarbons could have stimulated the activity of a small subset of the microbial community that was genetically equipped to metabolize these compounds. However, the net effect of light exposure on the microbes in the organic layer was suppression (relative to dark-controls), likely resulting from the degradation of the much larger pool of tannin-like formulas that the microbes were consuming (Fig. 1). Others have reported that photo-degradation of DOC in northern humic lakes suppressed microbial activity despite production of labile DOC readily consumed by microbes⁴⁰. Together, findings from this and previous studies suggest that the net effect of photo-altered DOC on microbial activity depends not only on the lability of the photo-products, but also on the types of DOC that the microbial communities were equipped to degrade prior to light exposure.”

Some other points that need clarifications:

1) The description of the inocula for the incubations is quite vague, presently are just described as "native" communities. I see from other publications that inoculum has been taken as the whole water from the same leachate filtered for DOC, indicate if this was also the case. How closely do these communities resemble those present in surface waters –e.g. how much does the composition change once the community is exposed to sunlight?

We used the 1.2 μm -filtered fraction of each soil water leachate as the inocula (as is described in the experimental design section in the methods). For example, for permafrost DOC we used the 1.2 μm -filtered fraction of soil water leached from permafrost soil. Each DOC treatment (light and dark control) received inoculum at 20% by volume.

Identifying the factors governing microbial community composition in the soils and surface waters around Toolik Lake Field Station has been a key objective of our research over the past ~15 years. Our findings have demonstrated that the structure of microbial communities in these soils and surface waters is governed by two principal factors. First, microbial communities in surface waters are structured by the inoculation of soil communities (Crump et al., 2013; ISME). That is, species richness is highest in soil waters and decreases with increasing residence time as water moves through streams and lakes in the fluvial network. Second, the species sorting that occurs in downslope rivers and lakes is in large part driven by the changes in DOC chemical composition (ref. 39). For example, shifts in the Toolik Lake microbial communities are synchronized with changes in DOC related to snow-melt or photo-alteration of DOC over the summer months. This relationship between DOC chemical composition and microbial community composition is consistently observed across pan-arctic watersheds (ref. 38). Given that the inocula used in this study was leached from soil organic matter, and that the composition of the community responded to changes in DOC chemical composition by sunlight,

we have no reason to assume that the community used in this study behaved differently than those in the surface waters draining the soils studied here.

2) The Methods never actually state the sample location and dates for the results shown in Fig. 1-3. As mentioned, Fig. 1 seems basically identical to Fig. 3 in ref 14 for the photo-produced and -degraded, suggesting the location is the same, Imnavait moist acidic tundra. Please state explicitly. Are all the results from a single leaching? Are the triplicate replicates from three separate experiments or are they measurement replicates conducted at the same time?

The methods section has been updated to clarify the sample location, dates, and experimental design (which are the same as reported on in references 14 and 22). The triplicate replicates are samples from three separate experiments, not measurement replicates conducted at the same time on one sample from one experiment. This point has been explicitly stated in the revised methods section.

“Thawed surface soil and frozen deep soil (permafrost) was sampled from three replicate pits in moist acidic tussock tundra within the Imnavait Creek watershed on the North Slope of Alaska (68.62°N, 149.29°W). Deionized water was added to soil from two layers of each pit, the annually-thawed, shallow organic layer (5-15 cm depth) and the deeper permafrost layer (95-105 cm), in order to generate the triplicate samples used in each experiment.²² The dissolved organic carbon fraction of the leachates was isolated using 0.45 µm filters (Geotech Environmental Equipment, Inc.)²². Organic and permafrost layer DOC from each pit was placed in UV-transparent Whirl-Pak bags (Nasco, Inc.) and exposed to natural sunlight for 24 hours alongside dark-controls^{13,14}. Light-exposed and dark-control DOC from each soil layer was incubated with an inoculum of native microbial communities at 6-7 °C. The inoculum was composed of leachate filtered through a Whatman GF/C (nominal pore size 1.2 µm) and added to comprise 20% of the sample volume.”

Methods:

3) The photochemical experiment was apparently conducted in natural sunlight (24 hours) for the Imnavait moist tundra, but a solar simulator Suntest XLS+ was used for the other four types. 18 hours in the simulator is described as comparable to 24 hours natural sunlight at the location which I find surprising. Although the spectra shown in Fig. S2 are similar, the solar spectrum looks like mid-day clear sky. The average over 24 h would be much lower (even in June, UV content is very low for the portion of the day the sun is near the horizon). The figure legend cites reference 3 as the source for the natural spectrum, but that reference does not have any data on solar irradiance. Given how small is the Suntest XLS+ exposure chamber the authors should explain how they were able to configure a 45 cm distance between the lamp and the sample tubes. The authors should keep in mind that borosilicate glass generally filters out the shortest wavelength part of the UVB (<315 nm), probably of minor importance in photobleaching but should be accounted for in any modeling calculations.

We added text to the methods and updated the supporting figure to compare the intensity of the simulated sunlight to natural sunlight in at Toolik Field Station. The figure legend has been

updated to indicate that the reference for calculating the natural sunlight spectrum is ref. 6 (Fig. S2) rather than ref. 3 (this was a typo, sorry).

The 45-cm distance referred to the distance between the top of the reflector parabola and the bottom of the chamber. The distance between the quartz glass and the bottom of the chamber is ~40 cm. This line has been removed from the methods and replaced with the comparison between the simulated and natural sunlight in figure S2.

Borosilicate vials were not used in this study, we fixed that typo in our methods section. All of the photo-exposures were conducted in Whirl-Pak bags which are more UV-transparent than borosilicate vials. We've measured the transparency of these bags in our previous work (ref 6, Figure S8) and have corrected for their less than 100% transmission in the UVB range in our previous work to calculate rates of photochemical degradation of DOC (ref 6).

“Filtered leachates (<0.45 μm) were transferred to UV-transparent Whirl-Pak bags (Nasco, Inc.) and exposed to 18 hours of simulated sunlight at 20 °C alongside dark-controls (Atlas Suntest XLS+). The simulated sunlight had a similar spectral shape, but was 2.4-fold more intense than average June sunlight at Toolik Field Station, AK (Fig. S2). Therefore, 18 hours of simulated sunlight is equivalent to approximately two days of natural light in June at the field station.”

Minor Comments

1) Abstract “experimental manipulations of soils”
Change "soils" to “soil leachates”

This change has been made.

2) Results – the first paragraph states some important findings, however every sentence has a reference to another report. I found it difficult to understand what was being presented as new data in this report and what was found previously.

The referencing in this paragraph has been revised to clarify the novel findings reported in this study vs. findings that were previously reported. In this study, we compared the formulas altered by sunlight (from ref. 14) to the formulas used by microbes (this study), and related these changes in DOC chemistry by sunlight and microbes to rates of microbial respiration in the light and dark (this study), and shifts in microbial community composition and gene expression (this study). The main discovery of this study is that the effect of sunlight depends on what the microbes were consuming prior to light exposure.

The following data are the new (not published before) components that led to the novel conclusions in this study:

- Bio-degraded formulas in Fig. 1 (and in Table S1, Fig. S1)*
- All data on the microbial response to photochemically altered DOC in Figs. 2, 3 (and in Table S2, Figs. S2, S4).*
- Data showing that the effects of sunlight on DOC composition applies to a wider range of soils than previously published by Ward & Cory 2016 (that is, the data in Tables S3 and*

S4 and Figure S3 are consistent with findings from Ward & Cory 2016, but these are new data from additional sites that have not been previously published).

We added a paragraph to the beginning of the Results and Discussion section to clarify the novel results from this study vs. results that were presented in previous studies:

“Here we explain how and why the photo-alteration of DOC draining the deep permafrost layer stimulates microbial activity¹³, but photo-alteration of DOC draining the shallow, annually-thawed organic layer suppresses microbial activity^{13,24}. We compare the chemical formulas altered by sunlight¹⁴ to the formulas used by microbes (Fig. 1), and relate these changes in DOC chemistry caused by sunlight and microbes to the rates of microbial activity in the light and dark (Fig. 2) and to the shifts in microbial gene expression and community composition (Fig. 3).”

3) “This explanation is unlikely because $22 \pm 1\%$ of the initial organic mat DOC could be classified as labile (aliphatic-like DOC with an O/C less than 0.5)²¹, which was larger than the $8 \pm 2\%$ of the DOC pool respired to CO₂ or fixed into biomass during the relatively short (5 d) incubation.”

In addition to the point about abundance already mentioned, I found it difficult to follow the reasoning here. The assimilation rate of a DOC class will depend more on its concentration than on its percentage, and the percent of the total DOC pool respired or fixed will depend on the abundance of bacteria as well as the lability of the DOC.

We agree that uptake of DOC by microbes depends on the concentration and lability of the compound taken up as well as bacterial abundance and community composition. The point we are trying to make here is that microbes took up allegedly “low lability” aromatic DOC in the organic layer despite the presence of aliphatic DOC expected to be “high lability”. We understand that our wording on the percentages of formulas and DOC removed was confusing, and we’ve revised this text to clarify our interpretation:

“There are three plausible explanations why microbes in the organic layer consumed aromatic-like DOC (i.e., lower lability), rather than the aliphatic-like DOC expected to be most labile. First, the concentration of higher-lability, aliphatic-like DOC in the organic layer may have been too low to sustain microbial populations. However, the concentration of aliphatic-like DOC in the organic layer was at least 600 $\mu\text{M C}$, about three to four times greater than the concentration of DOC respired to CO₂ or fixed into biomass during the relatively short (5 d) incubation (Table S3; based on ¹³C-NMR and mass spectrometry showing that 20-30 % of the DOC in the organic layer was aliphatic-like, defined as aliphatic DOC resonating from 0-60 ppm or aliphatic C with an O/C < 0.5)^{29,30}. These data suggest that microbial use of aliphatic-like DOC in the organic layer may have been limited by the lability, not the concentration, of this DOC. The second explanation for lack of consumption of aliphatic DOC by microbes native to the organic layer is that this DOC may have lower lability than expected. Each molecular formula detected by high-resolution mass spectrometry likely corresponds to many different structural- and stereo-isomers with distinct bonding environments and activation energies for microbes to overcome during metabolism³⁶. Thus, there might be critical differences in structure between the aliphatic-like DOC in the organic layer vs. that in permafrost, thereby limiting

microbial consumption of this C. Third, microbes with the metabolic capacity to degrade aliphatic-like DOC were either absent or too rare in the initial organic layer community to degrade this DOC, especially given the tight coupling between microbial community composition and the degradation of specific types of DOC^{23,24,37-39}. The relative likelihood of these three explanations can be tested by examining shifts in microbial activities, community compositions, and gene expressions in response to photochemically-altered DOC.”

4) “For example, 26% of photo-produced permafrost DOC formulas had the same exact masses as formulas degraded by permafrost microbes in the dark, and the remaining photo-produced formulas had labile characteristics (i.e., aliphatic-like; Fig. 1A; Table S1). “

and
“

For example, 39% of formulas removed by sunlight had the same exact masses as formulas consumed by microbes in the dark, and there was strong overlap in the composition of formulas removed by sunlight and formulas consumed by microbes (Fig. 1B, Table S1).”

These are important points, but neither Fig. 1 nor Table S1 show the formulas that were both degraded by microbes and photo- produced/removed. In the figure it looks like each formula can have only one symbol type.

We agree that these data will help the reader understand how we arrived at these important points that sunlight produced formulas labile to permafrost microbes, but degraded formulas labile to organic layer microbes. We updated the supporting information with a van Krevelen plot that shows overlap in the exact masses of formulas degraded by microbes and formulas photo-produced and removed (plotted for each permafrost and organic layer DOC in Fig. S1 Bottom).

5) The description of community change seems to be only in terms of Unifrac distance and there is no interpretation of the directional change in the MDS space. Suggest either explaining what are the axes and what the directions mean or just presenting the distances in an unscaled diagram.

In any case, I am not sure what new information is being added. If activity of a community is stimulated, composition change would be expected since some strains will be better able to use the available material than others. The significance of the overlap in the dark treatment at four hours and the light treatment at 7 days is not clear since the dark treatment has also shifted by about that much distance in 7 days.

Yes, we agree that the main point of Fig. 3 is consistent with the expectation that if a community is stimulated its composition will change because some strains are better suited to use the DOC than others. Our findings directly demonstrate this expectation (coupling the community shift to an abundance of DOC from the mass spec and NMR results, which as far as we know hasn't been done) by showing that the light vs. dark magnitude in the community shift is related to the light vs. dark abundance of DOC used by the microbes (Fig. 3C). We plotted that magnitude change in community composition (the weighted UniFrac distance) vs. abundance of DOC formulas used by microbes to highlight this change, and used the multidimensional scaling

diagrams to be clear about the direction of change in these communities. The figure showing magnitude of change (Fig. 3C) showed that there was a bigger shift in community composition for microbes fed light-exposed permafrost DOC compared to the same DOC kept in the dark, which we interpret is due to the production of labile C during light exposure (red points in Fig 3C). We used the multidimensional scaling diagrams (Figs. 3A and 3B) to make it clear that these communities did not change in the same direction. On its own, fig. 3C would be misleading and would suggest that the organic layer community in the dark controls changed in the same direction as the community in the light treatments. In fact, these communities changed in different directions, likely reflecting the differences in the chemistry of organic matter available for them to consume. There are two points we are making with figure 3 – one is about the magnitude of change related to % DOC change, and the other is about the direction of change: (1) a larger magnitude of change (i.e., greater distance on the plot and greater change in community composition) reflects the degree to which DOC supports the growth of the community, and (2) the direction of change reflects the differential adaptation of communities to the chemistry of the organic matter they are consuming. We revised the text pages 9-11 to clarify these two main points. In the revised submission we also modified Figure 3 such that 3C is now 3A, 3A is now 3B, and 3B is now 3C, which follows the logic of our explanation that the two points we are making include the magnitude and then the direction of change, and matches the topic order of the two revised paragraphs (included below):

“Photochemical production or removal of formulas fueling native microbes in the dark caused microbial communities to change in both magnitude and direction. The magnitude of community composition change from the initial inoculum in permafrost and organic layer DOC incubations was highest for the treatments that had higher growth rates (i.e., light-exposed permafrost DOC and dark-control organic layer DOC; Fig. 3A). In treatments with lower growth rates (i.e., dark-control permafrost DOC and light-exposed organic layer DOC), the communities changed less over time (Fig. 3A). This suggests that changes in microbial community composition depend on the abundance of DOC that the communities were equipped to degrade. In the permafrost incubations, the large shift in microbial community composition after four hours, especially for microbes incubated with light-exposed DOC (Fig. 3B), likely reflected the rapid growth of taxa that were adapted to consume newly-produced DOC (aliphatic-like DOC produced by light; Fig. 1A). This consumption of labile DOC is consistent with the higher rates of respiration in the light-exposed treatment, where photo-exposure increased the abundance of labile formulas (Fig. 1A) compared to the dark-control (Fig. 2). In the organic layer incubations, shifts in community composition over time were greater for microbes incubated with dark-control DOC (Fig. 3C), likely because there was no shortage of aromatic-like DOC that microbes were equipped to metabolize (Fig. 1B). In contrast, for microbes incubated with light-exposed DOC from the organic layer, the much smaller shift in community composition may be explained by the photo-removal of the aromatic-like DOC most used by this community (Fig. 1B).

The directions of change in community composition were different for permafrost and organic layer communities and for light and dark treatments (directions of arrows in Fig. 3B vs. 3C). In the permafrost layer incubations, after seven days the microbial community incubated with light-exposed DOC shifted to a community that resembled the dark-control community at four hours (Fig. 3B), likely reflecting the depletion of sunlight-produced DOC and the growth of taxa that were genetically adapted to consume the less labile dark-control DOC. This result suggests

strong overlap in the composition of DOC consumed by microbes between the dark treatment at four hours and the DOC remaining in the light treatment after seven days, because DOC chemistry is a principle control on microbial community composition^{23,24,37-39}. Such an overlap in composition is consistent with our suggestion that photo-exposure of permafrost DOC increased the abundance of compounds fueling microbes in the dark. In the organic layer incubations, after five days the community in the light-exposed treatment had changed little and, unlike in the permafrost incubations, this change was to a community that was different than the dark-control (Fig. 3C), suggesting that sunlight removed the compounds that were consumed by the dark-control community (Fig. 1B). These shifts in community composition are consistent with the reduction in activity levels in the light-exposed treatment (Fig. 2), and suggest that populations with different metabolic potentials must be required to degrade the new, photo-produced aliphatic-like DOC. Overall, the magnitude and direction of these shifts in permafrost and organic layers likely reflect the adaptation of microbial communities to the chemistry of the organic matter they are consuming, both over the short term (hours) due to metabolic (physiological) responses of individual cells, and over the long term (days, as shown in the 16S results of Fig. 3) due to selection for microbial populations (species) with better-suited metabolic machinery.”

6) “Thus, the negative effect of light on microbial processing of organic mat DOC may be temporary given that aquatic microbial communities can adapt to changes in DOC chemical composition on timescales of hours to days (Fig. 3)^{13,22,23,41}.”

However, the composition of the organic mat community in the light treatment did not change over 5 days.

The point we were trying make is that over time scales longer than our incubation (i.e., weeks), we expect that the light community from the organic layer may catch up in activity rates to the dark community. This expectation is based on results from longer-term incubation studies of DOC leached from the organic layer (e.g., ref. 24). In these earlier studies, we observed a lag effect in which the suppression of microbial activity by sunlight lessened with increasing incubation time. For example, in ref. 24 changes to DOC chemistry by sunlight suppressed microbial production by 50-70% over timescales of hours. However, over timescales of weeks, microbial production in the light-exposure community equaled or exceeded production in the dark-control community. Our hypothesis has been that this lag effect is related to the time required by the microbial community to adapt to the changes in DOC by sunlight (explained in the main text). The main text on page 12 has been updated to clarify that we expect the lag effect to operate on timescales of weeks, rather than hours to days.

7) " However, addition of hydrogen peroxide, a harmful ROS produced by DOC in sunlit surface waters, at a greater concentration than was produced during photo-degradation of permafrost and organic mat DOC¹⁴, showed no effect on microbial growth³⁹."

What was the concentration of the added H₂O₂ and where is it shown that there was no effect? Ref. 39 deals with Lake Superior bacterioplankton.

In reference 42, 0.7 μM H_2O_2 was added to Suwannee River fulvic acid, a reference DOC (representing 'terrestrially' derived DOC similar in composition to DOC in arctic freshwaters as shown in ref. 9). This DOC was then inoculated with Lake Superior bacterioplankton and incubated for a few days. The results in Fig. 5 in ref. 42 show that addition of H_2O_2 had no effect on the growth rate of the bacteria on Suwannee River DOC compared to the control (no H_2O_2 added). We revised the text to clarify how we think the findings from this paper support our interpretation of the results in this study. Our point is that harmful effects on microbes from reactive oxygen species (ROS), e.g., H_2O_2 , likely were not the main control on the microbial responses to photo-altered DOC. In large part this may be due to our experimental design, which minimized the effects of ROS on microbes for two reasons. First, the soil leachates were filtered prior to light-exposure. Following light-exposure, the leachates were inoculated with microbes. Therefore, by separating the light-exposure step from the inoculation step in time, we likely minimized the exposure of the microbial community to photochemically produced ROS because of the general fast reactivity and short residence time in water of ROS. Second, after photo-exposure, the leachates were placed in the dark at 4°C for 12 hours prior to inoculation. Given that DOC and trace constituents like iron are key sinks of ROS in the waters draining the soils studied here (Page et al., 2014), the ROS produced by light likely decayed to fairly low levels prior to inoculation, thereby minimizing the impact of ROS on the microbial community. Of course, in sunlit surface waters, light exposure probably alters microbial communities and their activities by affecting both DOC composition and by producing ROS, like H_2O_2 . We revised the text to clarify that in this study, what we are most likely detecting is the effect of light on DOC composition and how this photo-alteration impacts rates of microbial activities.

“An alternative to the explanation that microbial communities need time to adapt to photo-altered organic layer DOC is that photochemically produced reactive oxygen species (ROS) that can be harmful to microbes suppressed the consumption of photo-altered DOC. Photochemical production of ROS has been proposed to account for the lag in microbial growth or respiration from photo-altered DOC given that ROS decay over time following light exposure⁴⁶. However, our experimental design likely minimized any direct effect from ROS on microbes because the light exposure portion of our experiment was conducted without microbes (Table S5), and there was a period of about 12 hours between light exposure and addition of inoculum to organic layer and permafrost DOC treatments. Thus, photochemically produced ROS in the light-exposed DOC may have decayed to dark-control levels prior to addition of the bacterial inoculum by reacting with DOC or other constituents in the soil waters⁴⁷. Furthermore, others have concluded that the effects of light exposure of DOC on microbial growth rates were more likely due to changes in DOC composition than to harmful effects of ROS on microbes⁴². While it is possible that photochemical production of ROS alters microbial communities and their activities, both our experimental design and evidence from the literature suggest that the primary control on the microbial response observed in this study is the photochemical alteration to DOC chemical composition.”

8) Methods- "UV exposure during our experiments killed bacteria and reduced production by >99% (Table S5)."

Production is not a definitive measure that the bacteria are killed, leucine incorporation in incubations of bacterioplankton has been observed to largely recover during the night even after substantial inhibition occurs during the day.

A similar comment was also made by reviewer #3. Therefore, we have modified the text to indicate that under both dark and light conditions, filtration reduced bacterial production by >90% (these data are now provided in Table S5). Thus, over the time-scale of our photo-exposure experiment, bacterial activity was very low in the 0.45 µm-filtered water compared to the unfiltered water, demonstrating that the changes in DOM composition during the photo-exposure was due to changes caused by light, not by bacteria.

“Results indicate that even GF/F filters reduced bacterial contamination (measured as bacterial production) consistently in dark minus control samples ($93 \pm 2\%$, mean \pm SD), and UV exposure in the light treatment plus filtering reduced production even more ($99 \pm 1\%$, mean \pm SD; Table S5).”

8) Figure 1 Legend – misspelling "catergorized". States that formulas produced by or degraded by sunlight are shown in ref 13, but Table S1 states that they are shown in ref 14.

This change has been made.

9) Table 4.S1 "All data previously reported in Ward and Cory, In Review."
Update the reference

This change has been made.

Respectfully submitted,
Patrick Neale
Edgewater, MD

Reviewer #2 (Remarks to the Author):

General Comments

The authors present an interesting study focused on carbon-cycle dynamics in waters draining Arctic permafrost landscapes in northern Alaska. Specifically, this work examines the effects of photo-chemical modification of DOC composition and microbial respiration, with implications for C release from soils and surface waters to the atmosphere. The authors build upon prior work linking the physical effects of photo-oxidation and biological effects of microbial degradation on DOC production and turnover. Their findings show that sunlight can have both positive or negative impacts on microbial activity and respiration depending on whether dominant DOM pool is produced or removed via photo-oxidation. The authors used a combination of complimentary methods to disentangle mechanisms and test hypotheses, including FTICR-MS (for DOM composition), microbial activity and composition, and gene expression over the course of a 5-7 day incubation. The FTICR-MS approach used appropriate methods, and

interpretation of results relied heavily on methods/results reported by Cory et al. (2013 PNAS). The manuscript is well written and advances of our understanding of C dynamics in this critical region. The FTICR-MS approach used appropriate methods, and interpretation of results relied heavily on methods/results reported by Cory et al. (2013 PNAS). Most of my comments are relatively minor below.

Specific comments

1. Title: While it's true that some pathways "stimulate respiration", other findings show a reduction in rates. Consider rephrasing title for accuracy.

Yes, some photo-degradation of surface soil water has been shown to depress microbial activity over the short term (e.g., our ref. 24), we are not aware of any study that has shown that photo-degradation of permafrost DOC reduces rates of respiration. All papers to date have shown that photo-exposure of permafrost DOC stimulates microbial respiration (this study; ref. 13).

2. Page 3, Paragraph 2: Omit "soils" after "DOC leached from permafrost". In the literature, "permafrost soils" can include both active layer and perennially frozen soils (see Hugelius et al. 2014).

This change has been made.

3. Page 4, First line – Since "active layer soils" are defined above, I suggest using "active layer" throughout the manuscript. Further, the active layer thaws and refreezes annually, and is only "thawed" a couple months out of the year.

This change has been made.

4. Page 4, Last line – "Organic-horizon" or "Organic layer" are preferred terminology compared to "Organic mat".

"Organic mat" was changed to "Organic layer" throughout the text.

5. Page 6, Last paragraph – I like the discussion DOM lability in this section, but it could use a bit more detail/context. Could you add a sentence briefly describing the approach used by Ward & Cory (2015) to categorize lability? Why was the incubation only run for 5-7 days (and why the variability of 5-7?).

The text has been modified to explicitly state that we defined "labile" DOC as aliphatic formulas with an O/C less than 0.5 (based on references 28 and 29 in the main text). The citation to Ward and Cory (2015) was based on the formulas detected in permafrost and organic layer DOC, not on the lability of these formulas to microbial degradation. This citation has been replaced with references 28 and 29.

"Table S3; based on ^{13}C -NMR and mass spectrometry showing that 20-30 % of the DOC in the organic layer was aliphatic-like, defined as aliphatic DOC resonating from 0-60 ppm or aliphatic C with an O/C < 0.5^{29,30}."

The experiments were designed to run the incubation long enough to detect a change in microbial activity, community composition, and DOC chemical composition, while minimizing the amount of time the DOC and microbes spend in a bottle (i.e., minimize the “bottle effects”). The longer an incubation runs, the greater the differences between the bottle and natural bacterial communities, turnover rates, and regeneration of DOM pools (ref. 26). We expect that these differences can be large and even random over long time periods, which is why we aim for short incubation times. The range of 5 and 7 days for organic layer and permafrost DOC, respectively, was simply a logistical constraint. We were constrained on the front end of the experiments by weather (sunny days for photo-exposure) and the back end by the time it takes to process samples at the completion of the incubation. The availability of our team to carry out all the sample processing and analysis (i.e., DNA extractions, DOC characterization, respiration and bacterial production measurements, etc.) dictated when the incubation could be stopped. That is, the experiments were not stopped on the same day (i.e., on day 5) because we couldn't process all the samples from the two different DOC sources on the same days, so we had to let the permafrost DOC incubate longer than the organic layer DOC. However, all experiments were standardized by time (per day) than by “incubation”. We added the following sentence on page 16 in the Experimental Design section of the Methods:

“The relatively short incubation times were chosen to ensure the detection of a change in DOC chemical composition, microbial activity, and microbial community composition, while minimizing the amount of time the DOC and microbes spent in a bottle (i.e., “bottle effects”).”

6. Page 13 – I recommend adding a sentence or two to describe how soils were sampled and how leachings were conducted.

The methods have been updated to clarify how the soils were sampled and how the leachates were produced. Please see the specific revisions to the text in the Methods section on page 15, which are reproduced above in response to reviewer #1, minor comment 2.

7. Figure 2 – What statistical test was used here to determine differences across means? I don't see data analyses described anywhere in Methods.

The figure legend has been updated to indicate that we used an unpaired t-test.

8. Figure 3 – While I recognize space is limited, there is little to no text that describes the multidimensional scaling method and details of the results. I recommend adding a sentence or two somewhere to cover this in more detail.

This comment is similar to the 5th minor comment by reviewer #1 and second to last comment by reviewer #3. In addition to revising the text to expand our interpretation of the microbial community composition results, the methods have been revised to further explain the multidimensional scaling analysis, including a reference for the Unifrac distance calculation. We also revised the Figure 3 legend to better describe the results. The revisions to the text on the description of results include the two paragraphs starting on page 9, also listed in the response to Reviewer #1 minor comment 5, and the revisions to the text about the methods are on page 17 in the Methods section and in the caption to Figure 3, and are reproduced below:

“Changes in microbial community composition during incubation with light-exposed or dark-control organic and permafrost layer DOC (Figures 3 and S4) were quantified using amplicon Illumina sequencing (MiSeq 2x150 bp paired-end) of the V4 region of bacterial 16S rRNA genes^{39,54}. Amplicon sequences were paired using make.contigs (MOTHUR v.1.32.1)⁵⁵, and converted to QIIME format with split.groups from MOTHUR and add_qiime_labels.py from QIIME⁵⁴. Sequences were quality filtered with an expected error rate of 0.5, dereplicated (derep_fulllength), and abundance sorted (sortbysize) using USEARCH (v.7.0.1001_i86linux64)⁵⁶. Singleton sequences were removed, and reads were clustered (cluster_otus) at 97% similarity. Chimeras were removed with the de novo chimera check inherent in the cluster_otus, and with reference-based chimera filtering (uchime_ref) using the Gold Database (www.genomesonline.org) as reference. Reads (including singletons) were subsequently mapped back to the operational taxonomic units (OTUs) using UPARSE (usearch_global). Taxonomy of the representative sequences was assigned in QIIME (assign_taxonomy.py) using the RDP classifier trained to the SILVA database (v.111 database clustered to 97% OTUs). Patterns in beta-diversity, calculated as weighted Unifrac distance⁵⁷ were based on a rarefied OTU table (3800 sequences per sample) and displayed in non-metric multidimensional scaling diagrams to show the magnitude and direction of change in microbial community composition.”

And in the caption for Figure 3:

“(A) The magnitude of community shift from inoculum to the 4-hour time points (Weighted Unifrac distance) plotted against the abundance of DOC most used by microbes native to each soil layer. DOC most used by microbes was defined following the results in Fig. 1 and Table S1: permafrost = aliphatic-like, O:C<0.5, MW<500 DA; organic layer = aromatic-like, O:C>0.6, MW>500 DA. Error bars indicate ± standard error of the mean and in some cases are smaller than the symbol (N = 2 or 3). Non-metric multidimensional scaling diagrams showing the magnitude and direction of change in microbial community composition (16S rRNA gene amplicon sequences) among inocula and incubations for (B) permafrost soil leachate, and (C) organic layer soil leachate based on beta-diversity calculations (weighted Unifrac distances for a rarefied dataset; 3800 sequences per sample). Note that in (B, permafrost) the dark, 4-hr symbols are hidden behind the light, 7-d symbols.”

Reviewer #3 (Remarks to the Author):

The authors Ward et al., present a manuscript titled “Photochemical alteration of dissolved organic carbon draining permafrost soils shifts microbial metabolic pathways and stimulates respiration”. Overall, the manuscript is well presented and written, but I feel the following concerns need to be addressed before publication.

Results and Discussion

In the first line the authors introduce the term “organic mat”. I am not sure exactly what this is and how it differs from active layer. There is a line in the methods that indicates it is the seasonally thawed layer. I would move this information to the results section as the reader sees

this first in the organization of the manuscript. Also, please distinguish the difference between organic mat and active layer, it may be a common term, but one which I have not encountered before.

Following Reviewer #2's suggestion, we replaced "organic mat" with the preferred and technically sound term "organic layer."

In addition to describing the differences between the permafrost and organic layer in the Introduction, we also explicitly stated the differences between these soil layers in the first sentence of the Results and Discussion section:

"Here we explain how and why the photo-alteration of DOC draining the deep permafrost layer stimulates microbial activity¹³, but photo-alteration of DOC draining the shallow, annually-thawed organic layer suppresses microbial activity^{13,24}."

In the Results and discussion it is sometimes hard to distinguish findings from this study versus references to the literature. For example, in the same sentence there is reference to a figure and other literature. This sometimes makes it difficult to decide if the results are novel to this study or verify results of other studies. I would try to split these with statements like, similar to whoever et al., we found..... Try to highlight the novel findings of this study.

This comment is similar to a comment made by Reviewer #1, minor point 2 (and please also see our response there). We added a paragraph to the beginning of the Results and Discussion section to clarify the novel results from this study vs. results that were presented in previous studies. The referencing has been revised for the first paragraph of the Results and Discussion, which was confusing in that we referenced some of our previous work on the chemical composition of permafrost and organic layer DOC when discussing the new findings in this study on the DOC used by microbes.

Carful with the interpretation of the data. The authors state "microbes adapt their metabolic machinery". However, the data also shows that there were taxonomic shifts in the community. Adapting their metabolic machinery implies that a single organism is shifting its own gene expression in response to an environmental change. Instead, your data seems to more strongly imply that the environmental change enriches different populations with different metabolic potentials. I think this is an important distinction that should be called out.

We agree, it is important to recognize that shifts in metabolism of cells over minutes to hours (as indicated by transcriptomics) and shifts in overall community metabolism over days due to shifts in the selective advantage of certain populations under the new conditions, are probably both occurring. Because we do not have single-cell (or species) measurements of metabolic shifts over very short time periods, we have revised our interpretations to clarify that while both of these "adaptations" likely occur, the data from shifts in 16s as shown in Figure 3 are only conclusive for shifts in community composition.

"Overall, the magnitude and direction of these shifts in permafrost and organic layers likely reflect the adaptation of microbial communities to the chemistry of the organic matter they are

consuming, both over the short term (hours) due to metabolic (physiological) responses of individual cells, and over the long term (days, as shown in the 16S results of Fig. 3) due to selection for microbial populations (species) with better-suited metabolic machinery.”

And

“The suggestion that microbes adapt their metabolic machinery in response to photo-altered DOC was tested using metatranscriptomic measurements. Four hours into the incubation the expression of genes coding for the degradation of aromatic molecules (KEGG Tier IV Category) was significantly lower for the organic layer community incubated with photo-altered DOC compared to the dark-control community, regardless of whether expression was normalized to all KEGG gene expression or to Metabolism gene expression (KEGG Tier II Category; paired t-test, $p \leq 0.05$)⁴¹. Moreover, 15 of the 16 differentially-expressed aromatic-degradation genes had lower expression in the light treatment than in the dark-control⁴¹. This result suggests that microbes incubated with photo-altered organic layer DOC re-tooled their metabolic machinery to degrade the labile, aliphatic-like DOC that was produced in the light (Fig. 1B). Consistent with this interpretation, prior to light exposure of DOC the metabolic pathways of microbes native to the organic layer were more focused on consuming lower lability aromatic-like DOC rather than the less abundant aliphatic-like DOC (Fig. 1B and Table S1). Over time, this initial metabolic response would lead to a competitive advantage for populations with the metabolic potential to degrade the aliphatic-like DOC that was produced by sunlight (Fig. 1B). Together, the differential gene expression and changes in community composition (Fig. 3B and C) suggest that sunlight exposure either produced (in the permafrost layer) or removed (in the organic layer) the abundant DOC that the microbial community was equipped to degrade, thereby inducing changes to key metabolic pathways used by the native microbial communities to consume DOC.”

Methods

As mentioned previously I think the introduction to the methods needs to be in the results to introduce terms and concepts to the reader early.

Please see our response to your first comment where we describe explicitly how we introduced the terms and concepts to the reader earlier.

I would suggest that always in the phrase “by always >90%” should be replaced by consistently

This change has been made to the text.

While I understand that there are limits on sampling and sequencing etc. I think there needs to be some discussion to the fact that transcriptomes were generated at 4 hours and respiration data at 5-7 days. Also why organic mats were incubated 5 days versus 7 days for permafrost. While I don't see why this would affect the overall conclusions, it seems rather arbitrary. If it was based on some type of data or measurement I would make this abundantly clear, or else add some caveats to the data interpretation.

We revised the text in the methods to justify the time points of our incubations. Microbial activity is expected to change rapidly in response to changes in DOC composition, which was the motivation for measuring the transcriptomes at four hours. Unfortunately, it is not possible to detect respiration (measured as DIC production and oxygen consumption in our study) between 0 and 4 hours. Rates of microbial respiration are slow at the in-situ surface water temperatures (6-7°C) where we conducted the incubations, meaning that the change in DIC or O₂ would have been < 1 μM, and our instruments can only detect changes in DIC or O₂ (relative to killed controls) > 1 μM. The range of 5 and 7 days was simply a logistical constraint. We were constrained on the front end of the experiments by weather and the back end by key personnel and workload. The weather (i.e., sunny vs. cloudy) dictated when the experiment was initiated, and the time to process all the samples at the end of the experiments by team members dictated when the experiment could be broken down. We added the following to the Methods to clarify why relatively short time periods were used.

“The relatively short incubation times were chosen to ensure the detection of a change in DOC chemical composition, microbial activity, and microbial community composition, while minimizing the amount of time the DOC and microbes spent in a bottle (i.e., “bottle effects”).”

Changes in microbial composition. There needs to be more information of the community composition data. There is no information on what OTU definition was used in the calculations. Also tools in Quiime Mothur and Usesrach is not sufficient. Which tools, default parameters? Please expand. Someone who wanted to recapitulate your study would have no way to do so with the information you have provided.

This comment is similar to the 5th minor comment by reviewer #1 and the 8th comment by reviewer #2. We have expanded our interpretation of the microbial community composition results, as well as provided a detailed description of the amplicon sequence analysis in the methods. Please see the specific revisions to the text in the Methods section on page 17, which are reproduced above in response to reviewer #2, comment 8.

I also could not find a statement as to the public availability of the datasets. Please make sure that the datasets are available in a public repository before publication.

A “data availability” statement was added to the end of the Methods section.

“Data presented in this study has been made publicly available online within the Arctic Long Term Ecological Research database.”

REVIEWERS' COMMENTS:

Reviewer #1 (Remarks to the Author):

Authors have extensively revised their article and the revisions have addressed all the major issues that reviewers described for the initial version. There are just a few minor items that need attention.

Line 172: "These data suggest that microbial use of aliphatic-like DOC in the organic layer may have been limited by the lability, not the concentration, of this DOC. The second explanation for lack of consumption of aliphatic DOC by microbes native to the organic layer is that this DOC may have lower lability than expected."

These two sentences seem to be saying similar things in slightly different ways, the first sentence having been added in revision. Rephrase to improve the flow.

MDS plots

The additional text in the materials and methods clarifies how sequence data was handled but what is still needed are specific details or a reference on how the non-metric multi-dimensional scaling was done. It would also be interesting to know if there was any general characterization that can be given to the inferred community components in each axis.

Spectral Comparison (Figure S2)

This figure now gives a much better explanation of the relationship between solar simulator and natural sunlight except that it is seemingly inconsistent to show a representative solar spectrum that is about the same magnitude (as well as spectral distribution) as the solar simulator in S2-a but that the solar simulator is 2.4 times the average irradiance in S2-b. This inconsistency could be resolved by adding to the caption that the spectrum in S2a is for "natural clear sky, mid-day June sunlight at Toolik Field Station, AK" as it was described in the materials and methods of the first version. This makes it more consistent that average (that includes all 24 hours of day) is about 2.4x lower than noon or the solar simulator. Note that y axis units should be $\mu\text{mol photons m}^{-2} \text{s}^{-1} \text{nm}^{-1}$

Reviewer #2 (Remarks to the Author):

The authors have done an excellent job in responding to and addressing all comments by the three reviewers. The manuscript is now greatly improved with respect to clarity and methodological/analytical details. I have no additional comments, and recommend that the manuscript now be accepted for publication.

Reviewer #3 (Remarks to the Author):

Overall, the manuscript is much improved and I do not see any major issues that need to be addressed.

My one comment would be that while the authors have added a data availability statement as to the data being available in the LTER database, this is not the normal repository for sequence data. I would strongly recommend the authors submit their sequence data to a repository such as the NCBI SRA or MGRAST. These are databases much more familiar to people that may have an interest in

using the sequence data in future studies.

Red italics = response to reviewer comment

Black italics = revisions to text

Reviewer #1 (Remarks to the Author):

Authors have extensively revised their article and the revisions have addressed all the major issues that reviewers described for the initial version. There are just a few minor items that need attention.

- 1) Line 172: “These data suggest that microbial use of aliphatic-like DOC in the organic layer may have been limited by the lability, not the concentration, of this DOC. The second explanation for lack of consumption of aliphatic DOC by microbes native to the organic layer is that this DOC may have lower lability than expected.”

These two sentences seem to be saying similar things in slightly different ways, the first sentence having been added in revision. Rephrase to improve the flow.

Thank you for pointing out the redundancy of our wording. The manuscript has been modified to improve the flow of the sentence:

“These data suggest that microbial use of aliphatic-like DOC in the organic layer was likely not limited by the concentration of this DOC. Alternatively, the second explanation for lack of consumption of aliphatic-like DOC by microbes native to the organic layer is that this DOC may have lower lability than expected.”

- 2) MDS plots

The additional text in the materials and methods clarifies how sequence data was handled but what is still needed are specific details or a reference on how the non-metric multi-dimensional scaling was done. It would also be interesting to know if there was any general characterization that can be given to the inferred community components in each axis.

We named the software and added a reference for the non-metric multi-dimensional scaling. The axes of MDS diagrams are sometimes regressed against ancillary data, but, unlike other ordination techniques (e.g., PCA), MDS does not include any other information to ordinate the points other than the distance matrix. Our annotation of the MDS diagrams shows how the axes dimensions are related to time of incubation, but can also be thought of as relationships to the other factors that change during the incubations. Our new sentence for the methods section reads:

“Patterns in beta-diversity, calculated as weighted Unifrac distance (Lozupone & Knight, 2005), were based on a rarefied OTU table (3800 sequences per sample) and displayed in non-metric multidimensional scaling diagrams (Clarke 1993) using PRIMER-E software (V 7.0) to show the magnitude and direction of change in microbial community composition.”

- 3) Spectral Comparison (Figure S2)

This figure now gives a much better explanation of the relationship between solar simulator and natural sunlight except that it is seemingly inconsistent to show a representative solar spectrum that is about the same magnitude (as well as spectral distribution) as the solar simulator in S2-a but that the solar simulator is 2.4 times the average irradiance in S2-b. This inconsistency could be resolved by adding to the caption that the spectrum in S2a is for “natural clear sky, mid-day June sunlight at Toolik Field Station, AK” as it was described in the materials and methods of the first version. This makes it more consistent that average (that includes all 24 hours of day) is about 2.4x lower than noon or the solar simulator. Note that yaxis units should be $\mu\text{mol photons m}^{-2} \text{ s}^{-1} \text{ nm}^{-1}$.

The figure description and units were modified following the reviewer’s suggestion.

Reviewer #2 (Remarks to the Author):

The authors have done an excellent job in responding to and addressing all comments by the three reviewers. The manuscript is now greatly improved with respect to clarity and methodological/analytical details. I have no additional comments, and recommend that the manuscript now be accepted for publication.

Reviewer #3 (Remarks to the Author):

Overall, the manuscript is much improved and I do not see any major issues that need to be addressed.

My one comment would be that while the authors have added a data availability statement as to the data being available in the LTER database, this is not the normal repository for sequence data. I would strongly recommend the authors submit their sequence data to a repository such as the NCBI SRA or MGRAST. These are databases much more familiar to people that may have an interest in using the sequence data in future studies.

We have added the following text to the data availability statement to indicate that all sequence data will be uploaded to either the NCBI GenBank database (16S) or the JGI IMG Gold database (Metatranscriptome sequences).

“16S rRNA gene amplicon sequences are deposited in the NCBI Sequence Read Archive (SRA) under the bioproject accession number PRJNA356108. Metatranscriptome sequences and assembled contigs are publicly available via IMG under GOLD study ID Gs0114298.”